# Highly efficient generation of isogenic pluripotent stem cell models using prime editing

Hanqin Li[1,2,3†], Oriol Busquets[3,4†], Yogendra Verma[1,3], Khaja Mohieddin Syed[1,3], Nitzan Kutnowski[1], Gabriella R Pangilinan[1,3], Luke A Gilbert[3,5,6,7], Helen S Bateup[1,3,8,9], Donald C Rio[1,3]*, Dirk Hockemeyer[1,2,3,8]*, Frank Soldner[3,4,10,11]*

[1]Department of Molecular and Cell Biology, University of California, Berkeley, Berkeley, United States; [2]Innovative Genomics Institute, University of California, Berkeley, Berkeley, United States; [3]Aligning Science Across Parkinson's (ASAP) Collaborative Research Network, Chevy Chase, United States; [4]Dominick P. Purpura Department of Neuroscience, Albert Einstein College of Medicine, The Bronx, United States; [5]Helen Diller Family Comprehensive Cancer Center, University of California, San Francisco, San Francisco, United States; [6]Department of Urology, University of California, San Francisco, San Francisco, United States; [7]Arc Institute, Palo Alto, United States; [8]Chan Zuckerberg Biohub, San Francisco, United States; [9]Helen Wills Neuroscience Institute, University of California, Berkeley, Berkeley, United States; [10]Department of Genetics, Albert Einstein College of Medicine, The Bronx, United States; [11]Ruth L. and David S. Gottesman Institute for Stem Cell and Regenerative Medicine Research, Albert Einstein College of Medicine, The Bronx, United States

*For correspondence:
don_rio@berkeley.edu (DCR);
hockemeyer@berkeley.edu (DH);
frank.soldner@einsteinmed.edu
(FS)

†These authors contributed
equally to this work

Reviewing Editor: Simón
Méndez-Ferrer, University of
Cambridge, United Kingdom

**Abstract** The recent development of prime editing (PE) genome engineering technologies has the potential to significantly simplify the generation of human pluripotent stem cell (hPSC)-based disease models. PE is a multicomponent editing system that uses a Cas9-nickase fused to a reverse transcriptase (nCas9-RT) and an extended PE guide RNA (pegRNA). Once reverse transcribed, the pegRNA extension functions as a repair template to introduce precise designer mutations at the target site. Here, we systematically compared the editing efficiencies of PE to conventional gene editing methods in hPSCs. This analysis revealed that PE is overall more efficient and precise than homology-directed repair of site-specific nuclease-induced double-strand breaks. Specifically, PE is more effective in generating heterozygous editing events to create autosomal dominant disease-associated mutations. By stably integrating the nCas9-RT into hPSCs we achieved editing efficiencies equal to those reported for cancer cells, suggesting that the expression of the PE components, rather than cell-intrinsic features, limit PE in hPSCs. To improve the efficiency of PE in hPSCs, we optimized the delivery modalities for the PE components. Delivery of the nCas9-RT as mRNA combined with synthetically generated, chemically-modified pegRNAs and nicking guide RNAs improved editing efficiencies up to 13-fold compared with transfecting the PE components as plasmids or ribonucleoprotein particles. Finally, we demonstrated that this mRNA-based delivery approach can be used repeatedly to yield editing efficiencies exceeding 60% and to correct or introduce familial mutations causing Parkinson's disease in hPSCs.

## Editor's evaluation

In this manuscript, Li et al., directly compare different editing strategies for human pluripotent stem cells and demonstrate that prime editing is more efficient and precise, compared with double strand break-based methods. They also confirm the suitability of prime editing for the introduction of different mutations related to Parkinson's disease as a model. In this process the authors noted a lower editing efficiency of human pluripotent stem cells, compared with tumour cell lines, and explored ways to improve it. Nucleofection of in vitro-transcribed mRNA-based delivery approach significantly increased the editing efficiency, without the need to select for targeted clones, and multiple rounds of mRNA-based prime editing could yield near complete editing of hPSCs, including disease-causing mutations. The proposed platform paves the way for future prime editing methods for hPSCs.

## Introduction

One of the current challenges of using human pluripotent stem cells (hPSCs) to model human diseases is to precisely and efficiently engineer the genome to introduce designer mutations (*Hockemeyer and Jaenisch, 2016*; *Soldner and Jaenisch, 2018*). Currently, the predominant approach in hPSCs is to induce targeted DNA double-strand breaks (DSBs) using highly active site-specific nucleases, such as the clustered regularly interspaced short palindromic repeats (CRISPR)/Cas9 system (*Cong et al., 2013*; *Ding et al., 2013*; *Jinek et al., 2012*; *Jinek et al., 2013*; *Mali et al., 2013*) or protein engineering platforms including zinc finger nucleases (*Hockemeyer et al., 2009*; *Soldner et al., 2011*; *Zou et al., 2009*) and transcription activator-like effector nucleases (TALEN) (*Boch et al., 2009*; *Cermak et al., 2011*; *Hockemeyer et al., 2011*). Such targeted DSBs have been shown to substantially increase genome editing efficiency over conventional homologous recombination. However, since nuclease-induced DSBs are in most contexts preferentially repaired by non-homologous end joining compared with homology-directed repair (HDR) mechanisms, DSB-mediated genome editing frequently generates undesirable compound heterozygous editing outcomes with one correctly targeted allele and insertion or deletion (indel) on the second allele, causing the disruption of the protein-coding sequence (*Cox et al., 2015*). Therefore, it has been challenging to generate disease-associated dominant mutations in a heterozygous setting. By contrast, PE has been shown to overcome this limitation in a wide variety of cell types, as it does not require a DSB but directly repairs a nicked DNA strand (*Anzalone et al., 2019*). PE is a multicomponent editing system composed of a Cas9-nickase fused to a reverse transcriptase (nCas9-RT) and an extended prime editing (PE) guide RNA (pegRNA) that is reverse transcribed and functions as a repair template at the target site (*Anzalone et al., 2019*). While successful PE has been previously demonstrated in hPSCs (*Chemello et al., 2021*; *Chen et al., 2021*; *Sürün et al., 2020*), it remains unclear whether this approach has the potential to facilitate the generation of hPSC-based disease models. Here, we systematically compare different genome editing methods and show that PE is overall more efficient and precise to introduce heterozygous point mutations into hPSCs. Furthermore, by optimizing the delivery modality of the PE components, we were able to establish a highly efficient genome editing platform for hPSCs. By comparing plasmid, ribonucleoprotein (RNP), and in vitro transcribed mRNA delivery, we found that nucleofecting nCas9-RT as mRNA combined with synthetically generated and chemically modified pegRNAs yielded editing efficiencies exceeding 60%, which is comparable to efficiencies observed in tumor cell lines (*Anzalone et al., 2019*; *Nelson et al., 2021*). Together, these data indicate that PE has the potential to become the preferred method for genome editing of hPSCs.

## Results

To evaluate the potential use of PE to genetically modify hPSCs, we directly compared editing outcomes of PE to established CRISPR/Cas9 and TALEN targeting approaches with the goal of introducing disease-relevant point mutations. Initially, we chose to target the leucine rich repeat kinase 2 (*LRRK2*) gene to introduce the G2019S (G6055A) mutation (*Gilks et al., 2005*), which is one of the most frequent pathogenic substitutions linked to Parkinson's disease (PD). This mutation is found in approximately 4% of dominantly inherited familial PD cases, in both heterozygous and homozygous forms, and around 1% of sporadic PD cases (*Healy et al., 2008*). To introduce the G2019S (G6055A)

**eLife digest** From muscles to nerves, our body is formed of many kinds of cells which can each respond slightly differently to the same harmful genetic changes. Understanding the exact relationship between mutations and cell-type specific function is essential to better grasp how conditions such as Parkinson's disease or amyotrophic lateral sclerosis progress and can be treated.

Stem cells could be an important tool in that effort, as they can be directed to mature into many cell types in the laboratory. Yet it remains difficult to precisely introduce disease-relevant mutations in these cells.

To remove this obstacle, Li et al. focused on prime editing, a cutting-edge 'search and replace' approach which can introduce new genetic information into a specific DNA sequence. However, it was unclear whether this technique could be used to efficiently create stem cell models of human diseases.

A first set of experiments showed that prime editing is superior to conventional approaches when generating mutated genes in stem cells. Li et al. then further improved the efficiency and precision of the method by tweaking how prime editing components are delivered into the cells. The refined approach could be harnessed to quickly generate large numbers of stem cells carrying mutations associated with Parkinson's disease; crucially, prime editing could then also be used to revert a mutated gene back to its healthy form.

The improved prime editing approach developed by Li et al. removes a major hurdle for scientists hoping to use stem cells to study genetic diseases. This could potentially help to unlock progress in how we understand and ultimately treat these conditions.

mutation into hPSCs, we generated plasmid-based CRISPR/Cas9, TALEN and PE reagents (without [PE2] or with secondary nicking guide RNAs [ngRNA; PE3]) using previously established optimized design and targeting procedures (*Figure 1A*; *Anzalone et al., 2019*; *Hockemeyer et al., 2011*; *Hsu et al., 2021*; *Soldner et al., 2011*; *Soldner et al., 2016*). Briefly, we co-electroporated the human embryonic stem cell (hESC) line WIBR3 with the respective genome engineering components and an enhanced green fluorescent protein (EGFP)-expressing plasmid to allow for the enrichment of transfected cells by fluorescence activated cell sorting (FACS). Genome editing outcomes were evaluated in the transfected bulk cell population (EGFP-positive) 72 hr after electroporation and in single cell-derived subclones. Next generation sequencing (NGS) of amplicons spanning the G2019S (G6055A) region indicated that 2–4% of alleles carried the correct G2019S (G6055A) substitution (*Figure 1B* and *Figure 1—figure supplement 1A*). While we observed roughly comparable editing efficiencies using CRISPR/Cas9, TALEN, and PE (with secondary ngRNA [PE3]) gene targeting, all HDR-based approaches generated a significantly higher number of undesired editing outcomes with 19.6 and 3.3% indels for CRISPR/Cas9 and TALEN, respectively, compared with less than 0.5% for PE (*Figure 1B* and *Figure 1—figure supplement 1A*). Genotyping of expanded single cell-derived clones showed higher efficiency in generating heterozygous correctly targeted cell lines using PE primarily due to a substantially higher number of compound heterozygous editing outcomes for CRISPR/Cas9 and TALEN targeting with the correctly inserted G6055A sequence variant on one allele and indels on the second allele (*Figure 1C*, as identified by restriction fragment length polymorphism [RFLP] and Sanger sequencing analysis). Together, these data indicate that PE is overall more efficient and substantially more precise in generating heterozygous mutations in hPSCs, as compared with traditional DSB-based genome engineering approaches.

To scale the PE-based genome editing approach and streamline the derivation of correctly modified single cell clones, we applied a recently established genome editing platform, which employs multiplex low cell number nucleofection, limited dilution, and NGS-dependent genotyping instead of the time-consuming and laborious FACS sorting and manual single cell expansion steps to isolate correctly edited hPSC lines (see *Figure 1—figure supplement 2* and Experimental procedures for details). While this approach results in slightly lower overall bulk editing efficiencies (as determined by NGS), most likely due to the lack of FACS-based enrichment of transfected cells, the substantially reduced number of cells required and the streamlined workflow allows for highly efficient, multiplexed generation of genome-edited hPSC lines in parallel in less than 4 weeks (*Figure 1—figure*

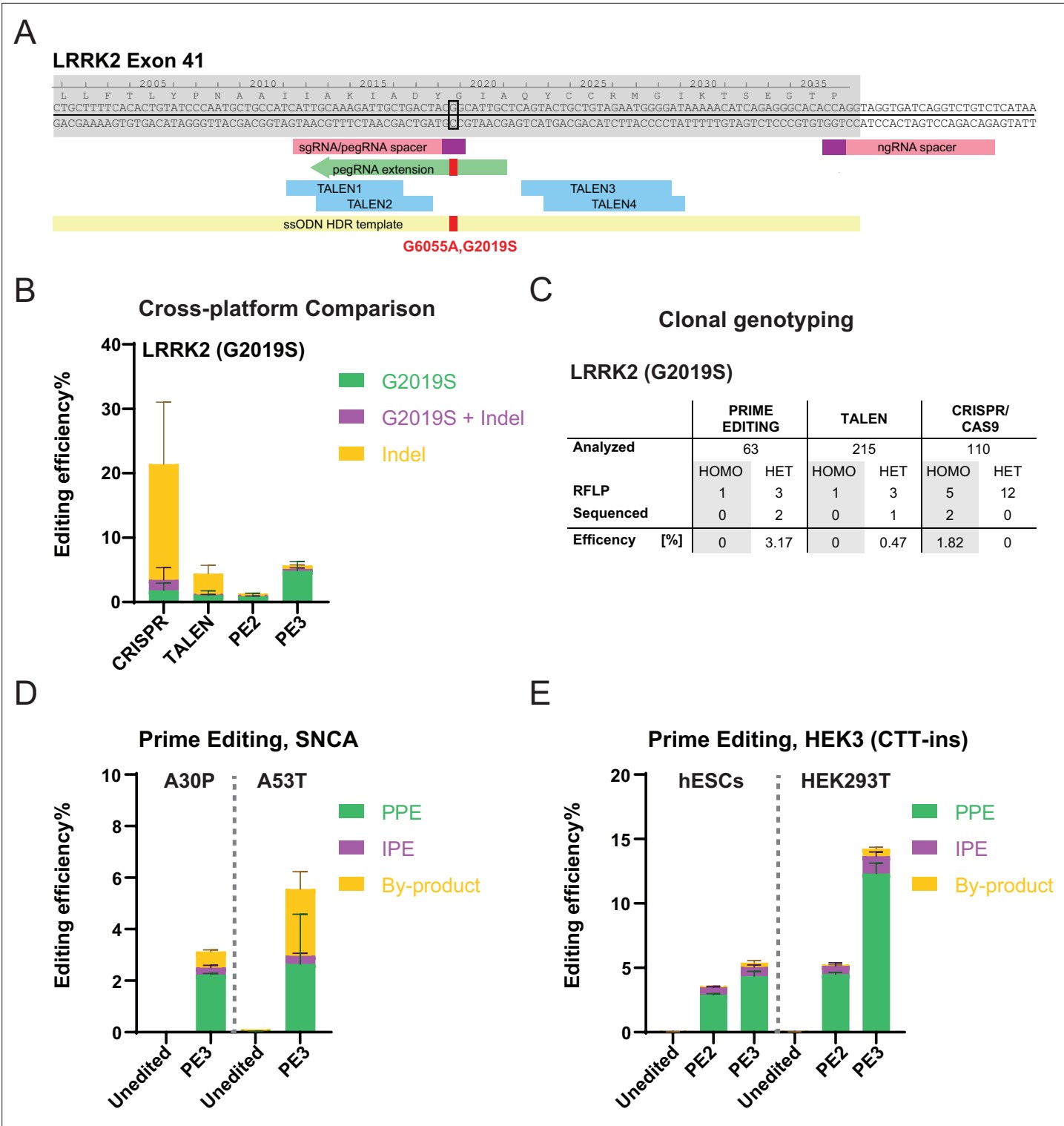

**Figure 1.** Systematic comparison of CRISPR/Cas9, transcription activator-like effector nucleases (TALEN), and prime editing (PE)-based genome editing efficiencies in human embryonic stem cells (hESCs) using plasmid-based delivery. (**A**) Schematic depicting genome editing strategies to generate the leucine rich repeat kinase 2 (*LRRK2*; G2019S[G6055A]) mutation. Exon, gray shade; prospacers for CRISPR/Cas9 and PE, pink boxes; protospacer adjacent motif (PAM) sequences, purple boxes; representative PE guide RNA (pegRNA) extension, green arrow; TALEN recognition sites, blue boxes; single-stranded oligodeoxyribonucleotide (ssODN) homology-directed repair (HDR) template for CRISPR/Cas9 targeting, yellow box; the nucleotide to mutate, black open square; intended mutation, red filled square. (**B**) Comparison of bulk genome editing outcomes between CRISPR/Cas9, TALEN, and PE to insert the *LRRK2* (G2019S) mutation (aggregated analysis using four different pegRNA designs and two TALEN pairs), N=2 for CRISPR/Cas9, N=3

*Figure 1 continued on next page*

*Figure 1 continued*

for TALEN, N=2 for PE2, and N=6 for PE3. For individual analysis see *Figure 1—figure supplement 1*. (**C**) A summary of clonal genotyping comparing different genome editing strategies indicating the number and editing efficiency of single cell-derived clones carrying the correct heterozygous (HET) or homozygous (HOMO) G2019S substitution as identified by restriction fragment length polymorphism analysis (RLFP) followed by Sanger sequencing to excluded additional insertions or deletions (indels). (**D**) Generating α-Synuclein (*SNCA*) familiar Parkinson's disease (PD) mutations, A30P, and A53T, using PE. pure prime editing (PPE), precise PE; IPE, impure PE; by-product, other indels. A30P, N=3; A53T, N=2. (**E**) Comparison of bulk PE outcomes on *HEK3* (CTT-insertion) edits between hESCs and HEK293T cells. Color scheme, same as (**D**). N=3. (Error bars indicate the SD, N=number of biological replicates).

The online version of this article includes the following figure supplement(s) for figure 1:

**Figure supplement 1.** Schematics of α-Synuclein (*SNCA*) Parkinson's disease (PD) mutation prime editing (PE) strategies and detailed leucine rich repeat kinase 2 (*LRRK2*) genome editing outcomes.

**Figure supplement 2.** A high throughput human pluripotent stem cells (hPSCs) genome editing pipeline combining limited dilution and next generation sequencing (NGS) genotyping.

**Figure supplement 3.** Genotyping and pluripotent marker staining of genome-edited human embryonic stem cells (hESCs) lines.

---

*supplement 2*). Importantly for this work, the omission of FACS enrichment allowed us to systematically and simultaneously compare a larger number of delivery modalities, as described below.

To confirm the feasibility of using PE to introduce point mutations efficiently and robustly into hPSCs, we tested PE at additional genomic loci. We were able to introduce mutations into the previously published and commonly targeted HEK site 3 (*HEK3*) locus (CTT-sequence insertion) (*Anzalone et al., 2019*), as well as two additional PD-associated mutations into the α-Synuclein (*SNCA*) locus (A30P [G88C]; *Krüger et al., 1998*; and A53T [G209A] *Polymeropoulos et al., 1997*; *Figure 1—figure supplement 1B, C*) with editing efficiencies comparable to the *LRRK2* locus (*Figure 1D and E*; quantified as pure prime editing efficiencies [PPE] as defined in *Petri et al., 2021*). As described for *LRRK2*, the analysis of single cell-expanded clones revealed the efficient generation of heterozygous and homozygous hPSC lines carrying the dominant A30P mutation in the *SNCA* gene (*Figure 1—figure supplement 3A, B*). Importantly, representative cytogenetic analysis of single cell-expanded *LRRK2* (G2019S) and *SNCA* (A30P) clones showed normal karyotypes for seven out of seven tested cell lines. Together, these experiments demonstrate that PE can be used to robustly and efficiently introduce disease-associated mutations into hPSCs to generate isogenic disease models.

During these experiments, we noted that editing outcomes for both the PE2 and PE3 approaches appeared considerably lower than what was previously reported for a variety of human tumor cell lines (*Anzalone et al., 2019*; *Nelson et al., 2021*). Indeed, we found that plasmid-based targeting of the *HEK3* (CTT insertion) locus resulted in only ~4.3% PPE in WIBR3 hESCs compared with ~12.7% PPE in HEK293T tumor cells using the PE3 strategy (*Figure 1E*). Similar differences in gene editing efficiencies between hPSCs, primary cells, and tumor cell lines have been commonly observed for other genome engineering approaches including CRISPR/Cas9 targeting (*Bowden et al., 2020*; *Haapaniemi et al., 2018*; *Ihry et al., 2018*). It remains unclear if this difference is the result of cell-intrinsic factors that restrict genome editing specifically in hPSCs or whether the low efficiency is a consequence of insufficient delivery of the PE components.

To estimate to which extend PE efficiencies could be increased in WIBR3 hESCs by optimized delivery of the prime editor, we used a recently described approach (*Bharucha et al., 2021*; *Habib et al., 2022*) and expressed the nCas9-RT protein (PE2 version of the prime editor protein as described in *Anzalone et al., 2019*) followed by a 2A-EGFP fluorescent reporter from the *AAVS1* safe harbor locus (*DeKelver et al., 2010*; *Hockemeyer et al., 2009*; *Figure 2A*). We established an hPSC clone that uniformly expressed GFP and maintained pluripotency (*Figure 2B* and *Figure 2—figure supplement 1*). Nucleofection of these cells with a plasmid encoding the *HEK3* (CTT insertion) pegRNA (without [PE2] or with [PE3] secondary ngRNA) or with a chemically-modified synthetic pegRNA (without [PE2] or with [PE3] secondary ngRNA) resulted in substantially increased editing efficiencies of up to 22% of correctly inserted modifications (*Figure 2C*). Similarly, targeting the *LRRK2* (G2019S) and *SNCA* (A30P) loci with chemically-modified synthetic pegRNAs (without [PE2] or with [PE3] secondary ngRNA) resulted in editing efficiencies up to 12% and 27%, respectively (*Figure 2D*). While these data do not exclude fundamental biological differences in the PE process between hPSCs and other cell types, these experiments demonstrate that the method of delivery of the PE components

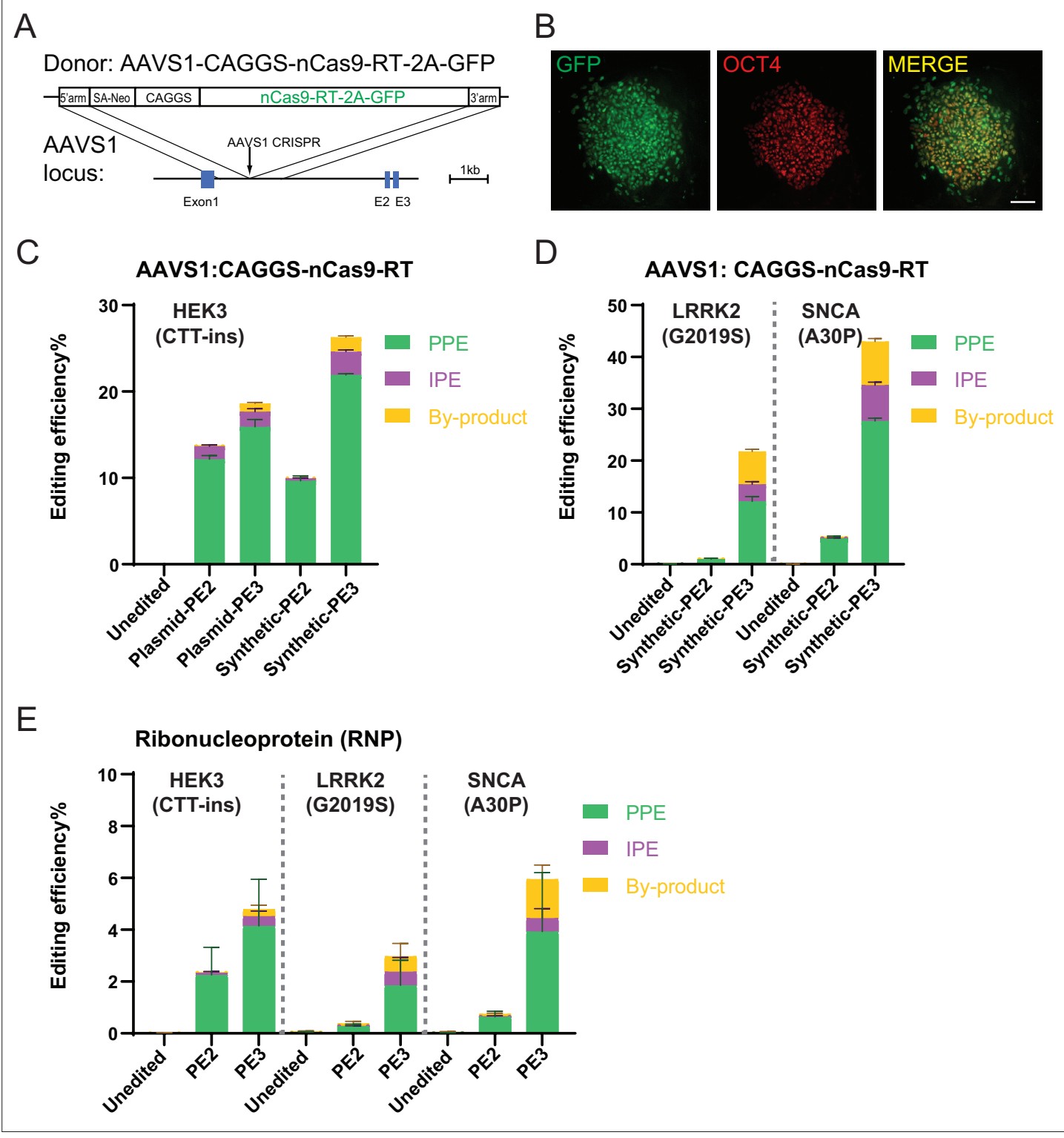

**Figure 2.** Prime editing (PE) in human embryonic stem cells (hESCs) expressing Cas9-nickase fused to a reverse transcriptase (nCas9-RT) protein from the *AAVS1* safe harbor or delivered as RNP. (**A**) Schematic of the genome editing strategy to knock-in nCas9-RT-2A-GFP (PE2 version of the prime editor protein as described in **Anzalone et al., 2019**) into the *AAVS1* locus. (**B**) Expression of green fluorescent protein (GFP) and immunostaining of OCT4 on hESCs with nCas9-RT-2A-GFP knock-in. Scale bar = 100 µm. (**C**) Comparison of bulk PE outcomes between plasmid-expressed PE guide RNAs (pegRNAs)/nicking guide RNAs (ngRNAs) and synthetic pegRNAs/ngRNAs on *HEK3* (CTT-insertion) edits in hESCs with nCas9-RT-2A-GFP knock-in. N=3. (**D**) Bulk PE outcomes on leucine rich repeat kinase 2 (*LRRK2*; G2019S) and α-Synuclein (*SNCA*; A30P) edits using synthetic pegRNAs/ngRNAs in

*Figure 2 continued on next page*

*Figure 2 continued*

hESCs with nCas9-RT-2A-GFP knock-in. N=3. (**E**) Bulk PE outcomes from RNP delivery on *HEK3* (CTT-insertion), *LRRK2* (G2019S), and *SNCA* (A30P) edits in human pluripotent stem cells (hPSCs). N=6. (Error bars indicate the SD, N=number of biological replicates).

The online version of this article includes the following source data and figure supplement(s) for figure 2:

**Figure supplement 1.** Pluripotent marker staining of human embryonic stem cells (hESCs) expressing Cas9-nickase fused to a reverse transcriptase (nCas9-RT) protein from the *AAVS1* safe harbor locus.

**Figure supplement 2.** Quality control and parameter testing of RNP-based prime editing (PE).

**Figure supplement 2—source data 1.** Unedited Coomassie blue staining gel.

has a significant role in dictating the genome editing efficiency in hPSCs, which can be comparable to the efficiencies observed in tumor cell lines and primary cells.

To improve PE efficiencies using transient delivery of the PE components, we set out to optimize PE delivery conditions. Initially, we focused on delivering the PE components as RNPs, a highly efficient approach described for CRISPR/Cas9-mediated genome editing (*Zuris et al., 2015*), which was recently successfully adapted for PE in zebrafish and human primary T cells (*Petri et al., 2021*). Using recombinant nCas9-RT protein (PE2 version of the prime editor protein as described in *Anzalone et al., 2019* purified from bacteria; *Figure 2—figure supplement 2A*) and the previously established protocols for RNP-based CRISPR/Cas9 editing (*Petri et al., 2021*; *Zuris et al., 2015*), we nucleofected pre-assembled RNPs containing the recombinant nCas9-RT protein and chemically modified synthetic pegRNAs (without [PE2] or with [PE3] secondary ngRNA) targeting the *HEK3* (CTT insertion), *LRRK2* (G2019S), and *SNCA* (A30P) loci. Consistent with previous reports on other cell types, we observed RNP-mediated editing outcomes in hPSCs with locus-dependent efficiencies between 1 and 6% (*Figure 2E* and *Figure 2—figure supplement 2B* and *Figure 2—figure supplement 2C*). While these data clearly indicate the feasibility of RNP-based PE in hPSCs, the observed efficiencies are comparable to the plasmid-based approach and far below the efficiencies observed with stable expression of nCas9-RT from the *AAVS1* locus. To exclude that RNP-based PE efficiencies were concentration- or batch-dependent, we repeated some of these experiments with protein from independently purified nCas9-RT batches (*Figure 2—figure supplement 2D*) and used higher protein concentrations (*Figure 2—figure supplement 2E*). However, none of these conditions resulted in substantially improved RNP-based editing efficiencies in hPSCs.

An alternative approach, allowing highly efficient delivery of Cas9 for CRISPR/Cas9-based genome editing (*Chang et al., 2013*; *Hwang et al., 2013*; *Wang et al., 2013*), is to deliver the prime editor using in vitro transcribed mRNA (*Chen et al., 2021*; *Sürün et al., 2020*). To systematically compare plasmid-, RNP-, and mRNA-based PE at the above-established *HEK3* (CTT insertion), *LRRK2* (G2019S), and *SNCA* (A30P) loci, we nucleofected either: (*i*) plasmid-based nCas9-RT together with plasmid-based pegRNAs (without [PE2] or with [PE3] secondary ngRNA), (*ii*) preassembled RNPs containing nCas9-RT protein and chemically modified synthetic pegRNAs (without [PE2] or with [PE3] secondary ngRNA), or (*iii*) in vitro transcribed nCas9-RT mRNA together with chemically modified synthetic pegRNAs (without [PE2] or with [PE3] secondary ngRNA). Bulk NGS revealed that the combination of in vitro transcribed mRNA-based delivery of the nCas9-RT with chemically modified synthetic pegRNAs and ngRNAs consistently increased editing efficiencies across all three tested loci up to 13-fold compared with plasmid- and 8-fold compared with RNP-delivery (*Figure 3A*). Using this optimized in vitro transcribed mRNA-based delivery approach allowed us to achieve editing efficiencies up to 26.7% (for the *SNCA* locus). When combined with secondary nicking of the non-edited strand (PE3), these editing efficiencies are similar to using the stably nCas9-RT expressing cell lines and comparable to efficiencies commonly observed in human tumor cell lines (*Anzalone et al., 2019*; *Nelson et al., 2021*). While mRNA-based editing efficiencies seem to depend on the approach used to in vitro transcribe the nCas9-RT mRNA (*Figure 3—figure supplement 1A*), we found that mRNA-based editing efficiencies are highly consistent across different nCas9-RT mRNA batches when using the best in vitro transcription conditions (*Figure 3—figure supplement 1B*). Furthermore, we observed that using single-stranded mRNA compared with either RNPs or double-stranded plasmid DNAs resulted in improved overall health and increased survival of single cells as indicated by increased clonal survival following nucleofection (*Figure 3B*). To test if the high efficiency of mRNA-based PE is a unique feature of the WIBR3 hESCs, we repeated some key experiments in a second human-induced pluripotent stem

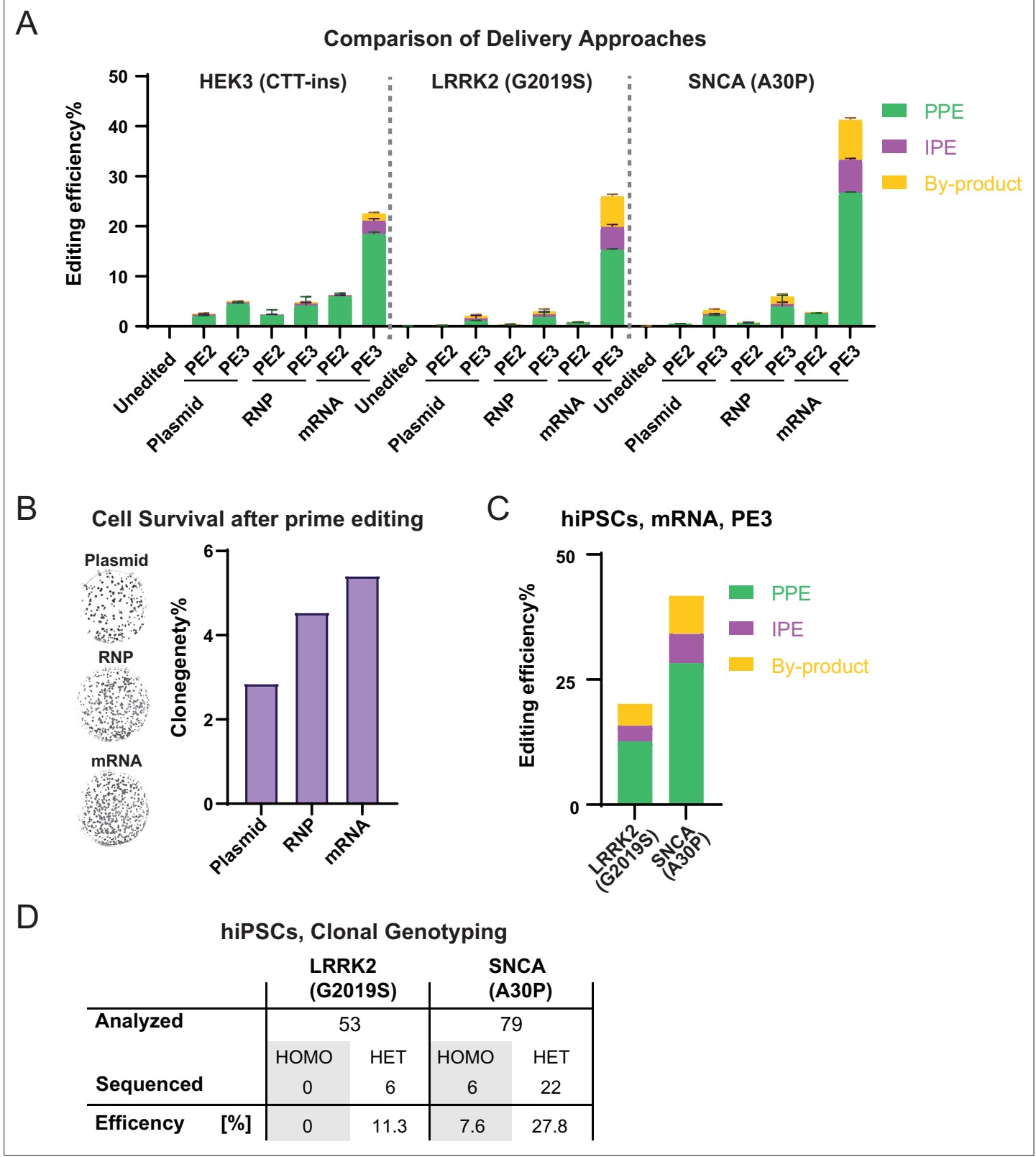

**Figure 3.** Highly efficient prime editing (PE) in human pluripotent stem cells (hPSCs) using mRNA-based delivery. (**A**) Comparison of bulk PE outcomes between plasmid, RNP, and mRNA-based delivery on indicated modifications in hESCs. Plasmid, mRNA groups, N=3; RNP data shown in *Figure 2E* was included in this analysis for direct comparison, N=6. (**B**) Representative images and quantification of alkaline phosphatase staining comparing clonogenicity of hESCs after nucleofection between plasmid, RNP, and mRNA-based delivery. N=2. (**C**) Bulk PE outcomes on leucine rich repeat kinase

*Figure 3 continued on next page*

*Figure 3 continued*

2 (*LRRK2*; G2019S) and α-Synuclein (*SNCA*; A30P) edits in a human-induced pluripotent stem cells (hiPSCs) line using mRNA-based delivery. N=2. (**D**) A summary of clonal genotyping from *LRRK2* (G2019S) and *SNCA* (A30P) PE in hiPSCs indicating the number and editing efficiency of single cell-derived clones carrying the correct heterozygous (HET) or homozygous (HOMO) substitution. (N=number of biological replicates, Error bars indicate the SD for samples N>2).

The online version of this article includes the following figure supplement(s) for figure 3:

**Figure supplement 1.** Quality control, parameter testing, and prime editing (PE) of feeder-free human embryonic stem cell (hESC) cultures with mRNA-based delivery.

cell (hiPSC) line 8858 (*Paşca et al., 2015*) by targeting the *LRRK2* (G2019S) and *SNCA* (A30P) loci and found comparable editing efficiencies (*Figure 3C*). Importantly, we were able to establish single cell-derived clones carrying the correct *SNCA* (A30P) and *LRRK2* (G2019S) mutations with high efficiency (*Figure 3D*). Considering that potential therapeutic applications would require precise genome editing of hPSCs in xeno-free conditions, we were able to show efficient and robust PE of the *LRRK2* (G2019S) locus in WIBR3 hESCs using several commonly used feeder-free culture systems with comparably high PE efficiencies (*Figure 3—figure supplement 1C*).

A major limitation of classical CRISPR/Cas9 targeting remains the high number and complexity of undesirable editing outcomes (indels). These alleles are resistant to targeting with the same reagents and thus limit the overall HR editing efficiencies in the context of continued editing or retargeting. Given the much reduced occurrence of indel-containing alleles in mRNA-based PE, we hypothesized that this approach might allow efficient retargeting of the same locus. Indeed, we find for all tested loci (*HEK3*, *LRRK2*, and *SNCA*) that additional rounds of mRNA-based PE of the same cell population could substantially increase overall editing efficiencies (*Figure 4A*). This indicates that multiple rounds of mRNA-based PE could result in precise and nearly complete editing of a bulk hPSC population without any type of selection.

The data presented thus far describes the insertion of disease-associated mutations into a wild-type genetic background. To test whether mRNA-based PE can be used to correct disease-causing mutations, we designed pegRNAs to specifically target only the mutated *SNCA* (A30P) allele to revert this mutation back to wild type in hPSCs. When targeting heterozygous (*Figure 4B and C*) or homozygous (*Figure 4D and E*) *SNCA* (A30P) hESC lines, bulk NGS indicated the correction of 31.0 (*Figure 4B*) and 30.1% (*Figure 4D*) of the mutated A30P alleles, respectively. Subsequent genotyping of single-cell derived clones indicated 26.2 (*Figure 4C*) and 48% (*Figure 4D*; combined heterozygous and homozygous) precisely corrected clones without additional undesired modifications (indels) of the wild-type allele. Taken together, our data indicate that in vitro transcribed mRNA-based PE is a highly efficient gene editing approach in hPSCs that has the potential to greatly facilitate the generation of disease-specific hPSC models.

## Discussion

The experiments performed here provide a detailed experimental road map for how to implement PE towards genome engineering of hPSCs. We show that mRNA transfection of the prime editor component (nCas9-RT) paired with the transfection of chemically modified guide RNAs is well tolerated and highly effective for introducing precise designer mutations in hiPSCs and hESCs. This work focuses on PE, which is highly versatile to introduce not only a wide range of disease-associated single nucleotide sequence variants but also more complex genetic alterations such as insertions and deletions. However, there are additional non-DSB-based genome editing approaches (e.g. based editors) which, dependent on the specific context, have been shown to efficiently introduce genetic modifications in hPSCs (we refer to recent reviews for a detailed discussion of the advantages and disadvantages of such approaches *Anzalone et al., 2020*; *Molla et al., 2021*; *Zeballos C and Gaj, 2021*). Although not tested, we believe that the here described RNA transfection-based delivery modalities could be adapted to increase genome editing efficiencies for genome editing approaches other than PE. Considering that mRNA-based PE does not require specialized molecular or biochemical skills and consistently achieves high editing efficiency in hESC and hiPSC lines, we predict that this approach has the potential to greatly facilitate the generation of disease-specific hPSC models and will be widely adopted by researchers.

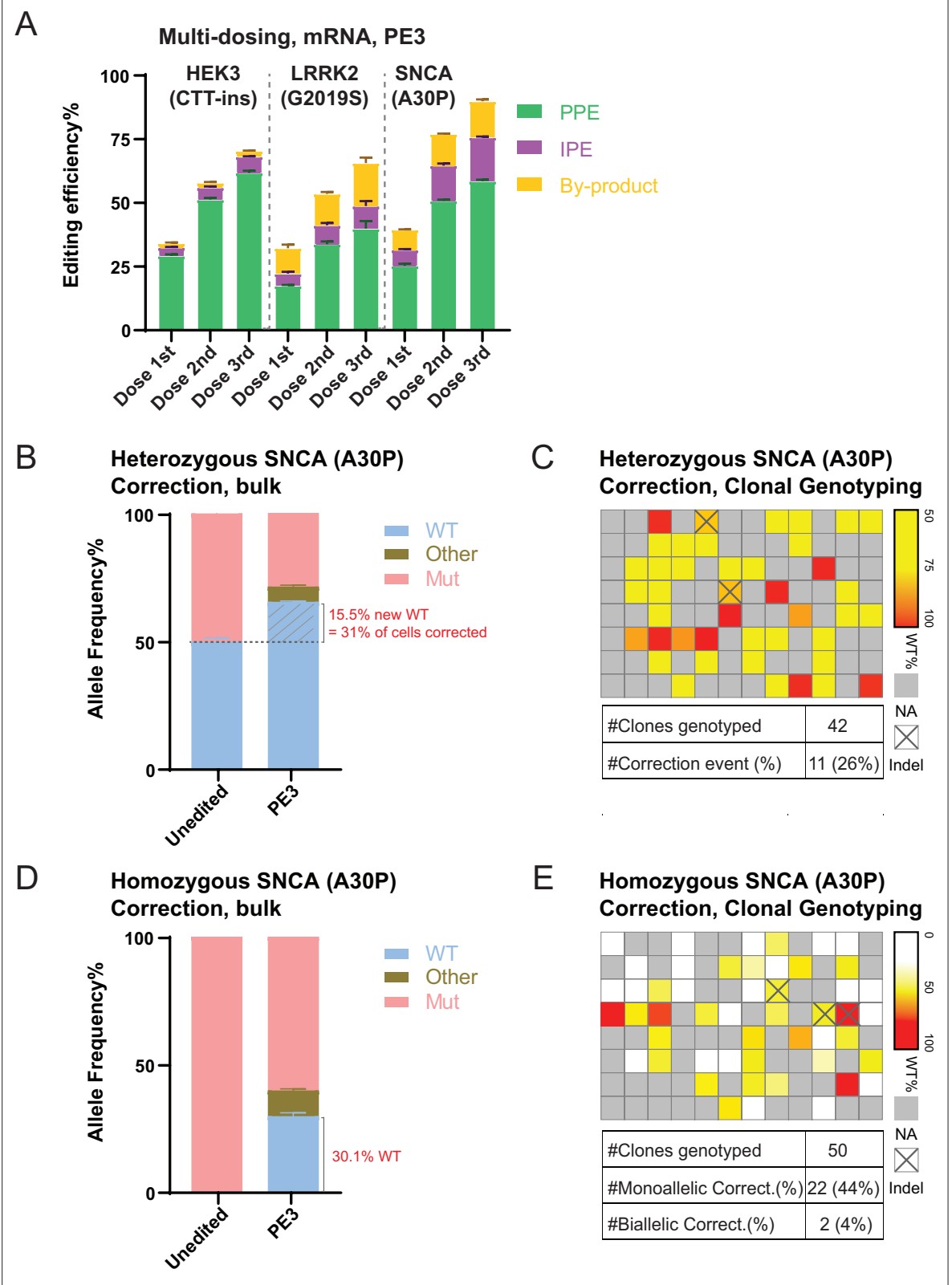

**Figure 4.** Repeated prime editing (PE) and reversion of an α-Synuclein (*SNCA*; A30P) mutation in human embryonic stem cells (hESCs). (**A**) Comparison of bulk PE outcomes in a multi-dosing strategy using mRNA-based delivery on the three indicated mutations in hESCs. N=3. (**B**) Bulk next generation sequencing (NGS) analysis indicating allele spectrum before (unedited) and after mRNA-based PE (PE3, single dosing) to correct the heterozygous *SNCA* (A30P) mutation in hESCs. N=3. The dashed line indicates the 50% allele frequency. The portion of the wild-type (WT) allele converted from the

*Figure 4 continued on next page*

*Figure 4 continued*

mutant allele is hatched. (**C**) Heatmap and summary of clonal genotyping from an *SNCA* (A30P) heterozygous mutation correction experiment in a 96-well format. The wells with edited clones containing more than 5% insertion or deletion (indel) reads were labeled as Indel. NA indicates wells without hESCs. (**D**) Bulk NGS analysis indicating allele spectrum before (unedited) and after mRNA-based PE (PE3, single dosing) to correct the homozygous *SNCA* (A30P) mutation in hESCs. N=3. (**E**) Heatmap and summary of clonal genotyping from an *SNCA* (A30P) homozygous mutation correction experiment in a 96-well format. The wells with edited clones containing more than 5% indel reads were labeled as Indel. NA indicates wells without hESCs. (Error bars indicate the SD, N=number of biological replicates).

During the process of establishing this workflow, we made several key observations. We find that PE can be as efficient in hPSCs as has been reported for cancer cells (*Anzalone et al., 2019*; *Nelson et al., 2021*). We demonstrate that this approach efficiently allows for the introduction or correction of heterozygous disease-related mutations in hPSCs with base-pair precision and without introducing undesired additional modifications on the second allele. The resulting cells showed a normal karyotype, consistent with low genotoxicity of PE due to the lack of DSBs (*Anzalone et al., 2019*).

A recent study reported comparable high PE efficiencies in a doxycycline-inducible PE2-expressing hESC line (*Habib et al., 2022*). The authors observe that PE3-mediated PE is generally accompanied by the generation of indels around the target site caused by the combinatory activity of the RT and pegRNA. While we also find a certain degree of indels (impure PE [IPE] and by-products) at the target site using the PE3 approach, the frequency is usually low compared with the intended sequence modifications which is consistent with previous data in other cell types (*Anzalone et al., 2019*; *Chen et al., 2021*; *Nelson et al., 2021*). While it has been shown that the frequency of undesired edits varies widely across genomic loci and is pegRNA dependent, it is possible that indel frequency is also affected by the different delivery modalities (e.g. different mRNA transfection of the PE component paired with the transfection of chemically modified guide RNAs has different kinetics and expression levels compared with other approaches). Due to the unique property of hPSCs to allow for the expansion of clonal cell lines, we do not believe that such on-target indels limit the use of PE for disease modeling approaches because undesired modifications can be easily excluded through targeted sequencing during quality control of individual clones.

In the past, generating such isogenic cell lines that differ exclusively at individual disease-causing sequence variants was highly laborious and an experimental bottleneck. Here we overcome this challenge by deploying PE via optimized delivery methods. We demonstrate that hPSCs can be subjected to several rounds of PE, eventually yielding up to 60% correctly targeted alleles. Importantly, PE efficiencies might be further increased by including mRNAs coding for DNA mismatch repair inhibiting proteins, a novel approach that has been recently shown to significantly improve the PE platform (*Chen et al., 2021*). These very high editing efficiencies without the need for selection of enrichment of targeted clones provide an intriguing platform to develop more robust in vitro disease models and potential therapeutic applications of PE in hPSCs or differentiated cell types. In our study, we successfully introduced three out of three familial PD point mutations into hPSCs using previously established algorithms to design pegRNAs (*Hsu et al., 2021*). In each case, a classical protospacer adjacent motif (PAM) was present close to the intended amino acid substitution and we did not explore more complex or challenging genetic modifications. As is the case for all genome editing approaches, PE efficiencies vary widely depending on the specific genomic context and pegRNA design, and certain genetic modifications will require more extensive pegRNA testing and validation. However, we expect systematic approaches that establish optimized design parameters for PE, as recently described for cancer cells (*Kim et al., 2020b*; *Nelson et al., 2021*) and the development of Cas9 variants with non-classical PAMs (*Chatterjee et al., 2020*; *Kleinstiver et al., 2015*; *Miller et al., 2020*) will overcome these limitations and combined with the optimized protocols reported here will allow PE to become a general method of choice for genome editing in hPSCs.

The focus of this work was to establish a highly efficient platform for PE in hPSCs and although all generated prime edited cell lines showed a normal karyotype consistent with low genotoxicity of PE due to the lack of DSBs (*Anzalone et al., 2019*), it is important to point out that we refrained from a detailed off-target analysis. We believe that there are already available substantial datasets in a variety of cell types indicating that PE is highly specific and shows much lower guide RNA-dependent or independent off-target effects compared with DSB-based CRISPR/Cas9 or base editing approaches (*Gao et al., 2021*; *Geurts et al., 2021*; *Habib et al., 2022*; *Jin et al., 2021*; *Kim et al., 2020a*; *Schene*

*et al., 2020*). Genome-wide off-target analyses of in vitro generated cell lines including whole genome sequencing-based approaches remain challenging even for conventional CRISPR/Cas9-based cutting approaches due to the substantial number of genetic alterations which occur during regular cell culture (*Kuijk et al., 2020*). Thus, a key future step toward the development of clinical PE approaches will require the development of sophisticated off-target analyses tools that account for nCas9 single strand break-mediated and transient reverse transcriptase expression-mediated genetic alterations.

## Methods
### hPSCs culture
All hESC and hiPSC lines were routinely maintained on irradiated or mitomycin C-inactivated mouse embryonic fibroblast (MEF) feeder layers as described previously (*Soldner et al., 2016*). Detailed protocols for culturing of MEFs and hPSCs can be found on protocols.io (https://doi.org/10.17504/protocols.io.b4msqu6e; https://doi.org/10.17504/protocols.io.b4pbqvin). The hiPSC 8858 line (Sergiu Pasca lab, Stanford; *Paşca et al., 2015*) and hESC line WIBR3 (NIH Registration Number: 0079; RRID:CVCL_9767; Whitehead Institute Center for Human Stem Cell Research, Cambridge, MA; *Lengner et al., 2010*) were maintained on MEFs in hESC media (Dulbecco's Modified Eagle Medium/ Nutrient Mixture F-12 (DMEM/F12) [Thermo Fisher Scientific]) supplemented with 15% fetal bovine serum (Hyclone), 5% KnockOut Serum Replacement (Thermo Fisher Scientific), 1 mM glutamine (Invitrogen), 1% nonessential amino acids (Thermo Fisher Scientific), 0.1 mM β-mercaptoethanol (Sigma) and 4 ng/mL fibroblast growth factor 2 (FGF2) (Thermo Fisher Scientific/Peprotech), 1× Penicillin-Streptomycin (Thermo Fisher Scientific). All cell lines have been routinely tested to be free of mycoplasma contaminations using a PCR-based mycoplasma detection test. Cultures were passaged every 5–7 days with collagenase type IV (Invitrogen; 1 mg/mL). The identities of all parental hESC and hiPSC lines were confirmed by DNA fingerprinting . If required (as indicated for the methods for the respective experiments), hiPSCs were adapted to feeder-free conditions on Vitronectin (VTN-N, Thermo Fisher Scientific) coated plates in mTeSR-plus (StemCell Technologies) or StemFlex (Thermo Fisher Scientific) media. Detailed protocols for feeder-free culturing of hPSCs can be found on protocols.io (https://doi.org/10.17504/protocols.io.b4mcqu2w).

### Culturing and transfection of HEK293T cells
HEK293T cells (RRID:CVCL_0063) were maintained in HEK293T media (DMEM [Thermo Fisher Scientific], 15% FB Essence [Avantor], 2 mM glutamine [Thermo Fisher Scientific], 1 mM nonessential amino acids [Thermo Fisher Scientific], 1× Penicillin-Streptomycin [Thermo Fisher Scientific]), and passaged every other day with 0.25% Trypsin with Ethylenediaminetetraacetic acid (EDTA) (Thermo Fisher Scientific). For transfection, cells were seeded into 0.2% gelatin-coated 12-well plates at $1 \times 10^4$ cell/cm$^2$. One day later, cells in each well were transfected with 500 ng pCMV-PE2 (pCMV-PE2 plasmid was a gift from David Liu. Addgene#132775; http://n2t.net/addgene:132775; RRID:Addgene_132775; *Anzalone et al., 2019*), 330 ng pU6-pegRNA and 170 ng pBPK1520-ngRNA for PE3 strategy or 500 ng pCMV-PE2, 500 ng pU6-pegRNA for PE2 strategy using 1 µL lipofectamine 2000 (Thermo Fisher Scientific) in opti-MEM (Thermo Fisher Scientific). Cells were collected for genomic DNA extraction and NGS-based allele quantification 3 days post-transfection. Detailed protocols can be found on protocols.io (https://doi.org/10.17504/protocols.io.eq2lynkzpvx9/v1).

### Molecular cloning
All standard molecular cloning procedures are performed following published protocols (*Green and Sambrook, 2012*). PegRNAs-expressing plasmids (pU6-pegRNA) were cloned by ligating annealed oligo pairs (*Supplementary file 2*) with BsaI-digested pU6-peg-GG-acceptor (pU6-pegRNA-GG-acceptor was a gift from David Liu. Addgene #132777; http://n2t.net/addgene:132777; RRID:Addgene_132777) as described previously (*Anzalone et al., 2019*). CRISPR-RNA expressing plasmids (px330-GFP) targeting the *LRRK2* locus were cloned by ligating annealed oligo pairs (*Supplementary file 2*) with BbsI-digested px330-GFP as described previously (*Soldner et al., 2016*). Nicking ngRNA-expressing plasmids (pBPK1520-ngRNA) were cloned by ligating annealed oligo pairs (*Supplementary file 2*) with BsmBI-digested pBPK1520 (BPK1520 was a gift from Keith Joung. Addgene#65777; http://n2t.net/addgene:65777; RRID:Addgene_65777) as described

previously (*Kleinstiver et al., 2015*). The heterodimeric TALEN pairs to target *LRRK2* (G2019S) were constructed and tested as described previously (*Cermak et al., 2011*; *Hockemeyer et al., 2011*) using the following variable di-residue (RVD)-containing tandem repeat sequences. Pair#1: TALEN1 (plasmid LRRK2-TALEN-TA01L): 5'-HD-NI-NG-NG-NN-HD-NI-NI-NI-NN-NI-NG-NG-NN-HD-NG-3' and TALEN3 (Plasmid LRRK2-TALEN-TA03R): 5'-HD-HD-HD-HD-NI-NG-NG-HD-NG-NI-HD-NI-NN-HD-NI-NN-NG-NI-HD-NG-3'. Pair#2: TALEN2 (Plasmid LRRK2-TALEN-TA04L): 5'-NN-HD-NI-NI-NI-NN-NI-NG-NG-NN-HD-NG-NN-NI-NG-3' and TALEN4 (Plasmid LRRK2-TALEN-TA07R): 5'- NI-NG-HD-HD-HD-HD-NI-NG-NG-HD-NG-NI-HD-NI-NN-HD-NI-NN-NG-3'.

## mRNA in vitro transcription

The plasmid pCMV-PE2 was cleaved with restriction endonuclease PmeI (100 µg DNA in 1 mL) for 4 hr at 37°C. The cleaved DNA was isolated by phenol-chloroform extraction and ethanol precipitation and resuspended at 500 µg/mL in TE buffer. The DNA was stored at –20°C. Eight 20 µL in vitro transcription reactions were set up using 1 µg of template DNA in each reaction using the New England Biolabs HiScribe T7 ARCA kit with tailing (E2060S; as per the manufacturer's instructions) and incubated for 2 hr at 37°C in an incubator (not a temp block). Using eight 20 µL reactions, after transcription, DNase I treatment, and polyA tailing (as per the manufacturer's instructions), the RNA was purified on four 50 µg New England Biolabs Monarch RNA cleanup columns (T2040L; as per the manufacturer's instructions) and eluted in 25 µL per column RNase-free $H_2O$ and pooled. The RNA was stored at –80°C and the yield from the total of eight reactions was ~200 µg purified PE2 mRNA by measuring $A_{260}$ on a Nanodrop 2000c spectrophotometer. Note that the nCas9-RT fusion mRNA is ~6500 nt. Detailed protocols can be found on protocols.io (https://doi.org/10.17504/protocols.io.b3fmqjk6).

## nCas9-RT protein purification

The nCas9-RT fragment from pCMV-PE2 was retrieved by BglII digestion and then cloned into the pET30a(+) expression vector (Novagen 69909) in the frame between the NotI and NdeI sites using NEBuilder HiFi DNA Assembly Master Mix (NEB) with bridging gblocks (*Supplementary file 1*) to encode a version of the protein bearing a C-terminal His6-tag. For protein expression, the plasmid was introduced into Rosetta 2 (pLysS). The cells were grown at 37°C and shaken at 175 rpm. Isopropyl β- d-1-thiogalactopyranoside (IPTG) (0.5 mM) was added at an OD600 of 0.6 and the cells were grown for 16 hr at 18°C. The cells were harvested by centrifugation (5000 × *g*) for 10 min at 4°C. Harvested cell pellets were washed with phosphate-buffered saline (PBS) and snap-frozen in liquid nitrogen for later purification. Cell pellets were thawed on ice, disrupted in 35 mL lysis buffer (25 mM HEPES-KOH pH 7.6, 1 M KCl, 20 mM imidazole, 1 mM dithiothreitol (DTT), 1 mM phenylmethylsulfonyl fluoride (PMSF), and 1 × protease inhibitor cocktail), briefly sonicated, then clarified by centrifugation at 25,000 × *g* for 30 min. Supernatants were filtered through a 0.22 µm syringe filter before application to 5 mL of nickel resin (Ni-NTA Superflow, QIAGEN) equilibrated in loading buffer (25 mM HEPES-KOH pH 7.6, 150 mM KCl, 20 mM imidazole, 1 mM DTT, and 1 mM PMSF). The resin was washed with 100 mL loading buffer followed by 50 mL wash buffer (25 mM HEPES-KOH, pH 7.6, 150 mM KCl, 40 mM imidazole, 1 mM DTT, and 1 mM PMSF). Protein was eluted in batch six times with 10 mL elution buffer (wash buffer + 500 mM imidazole). The eluted protein was diluted into a low-salt buffer (25 mM HEPES-KOH pH 7.6, 100 mM KCl, 1 mM DTT, and 1 mM PMSF), then loaded onto a 1 mL HiTrap heparin HP column (GE Healthcare) pre-equilibrated in low-salt buffer and eluted with a linear gradient of 100 mM to 1 M KCl over 40 column volumes. Peak fractions were concentrated to 8 mg/mL using a Spin-X UF 20 50 kDa molecular weight cutoff (MWCO) (Corning). Protein concentration was determined by UV absorbance at a wavelength of 280 nm. The final protein purity was determined by sodium dodecyl sulfate–polyacrylamide gel electrophoresis (SDS–PAGE) and Coomassie staining to be around 90%. The C-terminal His6-tag was not removed prior to the experiments. Detailed protocols can be found on protocols.io (https://doi.org/10.17504/protocols.io.b4yxqxxn).

## PE, CRISPR/Cas9, and TALEN-based genome editing using plasmid vectors

As indicated for the respective experiments, plasmid vector-based PE was performed using electroporation or nucleofection (using the high throughput hPSCs genome editing pipeline described below). Detailed protocols can be found on protocols.io (https://doi.org/10.17504/protocols.io.b4qnqvve).

Briefly for electroporation-based plasmid-mediated TALEN, CRISPR/Cas9, and PE, hPSCs were cultured on MEFs in Rho-associated protein kinase (ROCK)-inhibitor (10 µM, Stemgent; Y-27632) for 24 hr before electroporation. Cells were harvested using 0.05% trypsin/EDTA solution (Thermo Fisher Scientific) and resuspended in PBS. $1 \times 10^7$ cells were electroporated (Gene Pulser Xcell System, Bio-Rad: 250 V, 500 mF, 0.4 cm cuvettes) with the following plasmid vectors: for the PE PE2 strategy, we used 33 µg pCMV-PE2-GFP (*Anzalone et al., 2019*), (pCMV-PE2-P2A-GFP was a gift from David Liu. Addgene#132776; http://n2t.net/addgene:132776; RRID:Addgene_132776) and 12 µg pU6-pegRNA. For the PE PE2 strategy, we used 33 µg pCMV-PE2-GFP, 12 µg pU6-pegRNA, and 5 µg pBPK1520-ngRNA. For TALEN editing, we used 7.5 µg for each (left and right) TALEN-nuclease plasmid, 10 µg pEGFP-N1 (Clontech, Takara Bio USA, 6085–1), and 26 µg ssODN (single strand oligonucleotide containing the respective modification). For CRISPR/Cas9 editing, we used 16 µg pX330-GFP guide RNA (gRNA) and 26 µg ssODN (single strand oligonucleotide containing the respective modification). A list of the respective plasmids can be found in *Supplementary file 2*. Cells were maintained on MEFs for 72 hr in the presence of ROCK-inhibitor followed by FACS sorting (FACS-Aria; BD-Biosciences) of a single-cell suspension. EGFP expressing cells were either directly used for bulk NGS-based allele quantification or subsequently plated at a low density on MEFs in hESC media supplemented with ROCK-inhibitor for the first 24 hr. Individual colonies were picked and expanded 10–14 days after electroporation. Correctly targeted clones were subsequently identified by RFLP and genomic sequencing (see *Supplementary file 1* for respective primer sequences).

For nucleofection-based PE, hPSCs were cultured on MEFs in ROCK-inhibitor for 24 hr before nucleofection. Cells were harvested using collagenase IV (Thermo Fisher Scientific) followed by accutase (Thermo Fisher Scientific) and $5 \times 10^5$ cell were resuspended in 20 µL nucleofection solution and nucleofected (4D-Nucleofector TM Core +X Unit [Lonza], nucleofection program P3 primary cell, CA137) using the following plasmids vectors: for the PE PE2 strategy, we used 500 ng pCMV-PE2 and 500 ng pU6-pegRNA. For the PE PE2 strategy, we used 500 ng pCMV-PE2, 330 ng pU6-pegRNA, and 170 ng pBPK1520-ngRNA. A list of the respective plasmids can be found in *Supplementary file 2*. After nucleofection, cells were maintained either on MEFs in hESC media or on VTN-N coated plates in feeder-free media, both containing ROCK-inhibitor and either used for NGS-based allele quantification or single cell cloning (following the high throughput hPSCs genome editing pipeline described below).

## PE using RNP

The hPSCs cultured on MEFs were harvested and nucleofected using the same procedure as described in the plasmid delivery section, except with RNPs consisting of 90 pmol purified nCas9-RT protein, 300 pmol chemically modified synthetic pegRNA (*Supplementary file 2*) for PE2 strategy or 90 pmol purified nCas9-RT protein, 200 pmol chemically modified synthetic pegRNA, and 100 pmol chemically modified synthetic ngRNA (*Supplementary file 2*) for PE3 strategy. For increased protein doses, 270 pmol purified nCas9-RT protein were used instead. All RNPs were pre-assembled at room temperature (RT) for 10 min before nucleofection. Detailed protocols can be found on protocols.io (https://doi.org/10.17504/protocols.io.b4qnqvve).

## PE using mRNA

The hPSCs cultured on MEFs were harvested and nucleofected using the same procedure as described in the plasmid delivery section, except with 4 µg in vitro transcribed nCas9-RT mRNA, 150 pmol chemically modified synthetic pegRNA for PE2 strategy, or 4 µg in vitro transcribed nCas9-RT mRNA, 100 pmol chemically modified synthetic pegRNA and 50 pmol chemically modified synthetic ngRNA for PE3 strategy (*Supplementary file 2*). For feeder-free culture, hPSCs were harvested using accutase. In multidosing experiments, after the first nucleofection, hPSCs were nucleofected for the second and third time on days 7 and 14, respectively. Detailed protocols can be found on protocols.io (https://doi.org/10.17504/protocols.io.b4qnqvve).

## Genotyping of single cell expanded genome edited hPSCs clones by RLFP

The RFLP analysis was performed as previously described *Hernandez et al., 2005* following standard protocols (*Green and Sambrook, 2012*) to screen single-cell expanded clones for the insertion of the *LRRK2* (G2019S) mutation. Genomic DNA was amplified with primers SP-LRRK2-RLFP and ASP-LRRK2-RLFP (*Supplementary file 1*) under standard PCR conditions followed by restriction digest with SfcI. The *LRRK2* (G6055A, G2019S) mutation at coding nucleotide 6055 of *LRRK2* creates a novel SfcI cleavage site which allows to distinguish the two alleles after separation on a 3% agarose gel with the reference allele generating fragments of 228 bp and 109 bp and the mutated allele generating fragments of 207 bp, 109 bp, and 21 bp. Mutation carrying clones were further analyzed using Sanger and NGS sequencing of the PCR product.

## High throughput hPSCs genome editing pipeline

After nucleofection, cells were directly seeded onto MEF 96-well plates, at seeding densities of 10 cells/well in hPSCs media containing ROCK-inhibitor. Media was changed on days 4, 7, 10, 12, and 13 and ROCK-inhibitor was supplemented on day 13. On day 14, cells were washed with PBS once and then treated with 40 µL 0.25% trypsin for 5 min at 37°C, then 60 µL hPSC media containing ROCK-inhibitor was added to each well to inactivate trypsin. Cells were then gently triturated and 50 µL cell suspension was transferred to a 96-well PCR plate preloaded with 50 µL 2 × lysis buffer (100 mM KCl, 4 mM MgCl$_2$, 0.9% NP-40, 0.9% Tween-20, 500 µg/mL proteinase K, in 20 mM Tris-HCl, pH 8) for DNA extraction. The remaining 50 µL of cells were reseeded to a new MEF 96-well plate preloaded with 100 µL hPSC media containing ROCK-inhibitor and cultured for another 7 days with hPSC media changed daily. Meanwhile, the lysed cells in 96-well plates were incubated at 50°C overnight and then heated to 95°C for 10 min to inactivate the proteinase K. A ~300 bp genomic region covering the designed mutation was amplified using primers (*Supplementary file 1*) containing NGS barcode attachment sites (GCTCTTCCGATCT) from 2 µL cell lysis from each well with Titan DNA polymerase. Amplicons were purified at the UC Berkeley DNA Sequencing Facility, then i5/i7 barcoded in indexing PCR, pooled, and sequenced on 150PE iSeq in the NGS core facility at the Innovative Genomics Institute. CRISPResso2 (*Clement et al., 2019*) in PE mode was used to analyze the NGS data to identify wells containing the designed mutation, with the following criteria. Heterozygous candidates: number of reads aligned >100, 70%> mutant allele frequency >20%, indels frequency <5%; homozygous candidates: number of reads aligned >100, mutant allele frequency >70% and indels frequency <5%. Cells in those identified wells were single-cell subcloned once to ensure clonality. Detailed protocols for high throughput hPSCs genome editing (https://doi.org/10.17504/protocols.io.b4mmqu46) and genotyping by NGS (https://doi.org/10.17504/protocols.io.b4n3qvgn) can be found on protocols.io.

## *AAVS1* locus knock-in

To clone the AAVS1-SA-neo-CAGGS-nCas9-RT-2A-GFP targeting vector, the PmeI/SacII digested nCas9-RT fragment of pCMV-PE2-GFP were Gibson assembled into the EcoRI/KpnI sites of a parental AAVS1-SA-neo-CAGGS vector (a gift from Dr John Boyle). The hPSCs cultured on MEFs were harvested and nucleofected as described in the plasmid delivery section, except with 1 µg targeting vector and pre-assembled RNP consisting of 80 pmol purified Cas9 (Macrolab, UC Berkely) and 300 pmol chemically-modified sgRNA (Synthego) targeting the *AAVS1* locus (*Supplementary file 1*). Cells were replated onto DR4 MEFs postnucleofection in hESC media containing ROCK-inhibitor then selected with 70 µg/mL G418 (invitrogen) for 10 days with media change daily from day 3. Survived clones were manually picked, expanded, gDNA extracted and PCR genotyped with primers (*Supplementary file 1*) flanking each homologous arms using PrimeStar GXL DNA polymerase (Takara). Correctly targeted clones were further expanded and banked. Detailed protocols can be found on protocols.io (https://doi.org/10.17504/protocols.io.b37kqrkw).

## Karyotyping using array-based comparative genomic hybridization (aCGH)

Human hPSCs cultured on MEFs were harvested using collagenase IV as big aggregates and settled 3 times in washing media (DMEM [Thermo Fisher Scientific], 5% Newborn Calf Serum [Sigma], 1× Penicillin-Streptomycin [Thermo Fisher Scientific]), then strained by an 80 µm strainer. Cell aggregates

that did not pass through the strainer were collected, snap frozen as cell pellet, then sent to Cell Line Genetics (Madison, WI) for aCGH karyotyping. Detailed protocols can be found on protocols.io (https://doi.org/10.17504/protocols.io.kxygxzdrov8j/v1).

## Pluripotent marker staining

For immunostaining, hPSCs cultured on MEFs were fixed in 4% paraformaldehyde at RT for 10 min, permeabilized in 0.3% Triton-X100/PBS for 20 min, blocked in blocking solution [3% bovine serum albumin (BSA/PBS)] for 1 hr, incubated with primary antibody (OCT4 [DSHB Cat# PCRP-POU5F1-1D2, RRID:AB_2618968], 1:200; SSEA4 [DSHB Cat# MC-813–70 [SSEA-4], RRID:AB_528477], 1:200) in blocking solution at 4°C overnight, then washed with PBS and incubated with secondary antibody (Thermo Fisher Scientific Cat# A-11001, RRID:AB_2534069, 1:1000) in blocking solution at RT for 1 hr. For alkaline phosphatase staining, cells were fixed with cold 4% paraformaldehyde for 10 min, equilibrated with 100 mM Tris-HCl, pH 9.5 for 10 min at RT, then incubated with nitro-blue tetrazolium and 5-bromo-4-chloro-3'-indolyphosphate (NBT/BCIP) (SK-5400, Vector laboratories) at RT for 2 hr to overnight. Images were acquired on a Zeiss Axio Observer A1 inverted fluorescence microscope. Detailed protocols can be found on protocols.io (https://doi.org/10.17504/protocols.io.b4yyqxxw).

## Single-cell survival assay

The hPSCs nucleofected with the plasmid, RNP, or mRNA were seeded to MEFs at 100 cells/cm$^2$, then cultured for 14 days with media changed every other day. Cells were then stained for alkaline phosphatase as described above and the number of colonies in each condition was counted. Detailed protocols can be found on protocols.io (https://doi.org/10.17504/protocols.io.4r3l2okxxv1y/v1).

## Bulk NGS and allele quantification

Edited bulk cells were collected using trypsin at day 5 postnucleofection, then DNA extracted, mutation-region amplified, NGS and analyzed as described above detailed protocols can be found on protocols.io (https://doi.org/10.17504/protocols.io.b4n3qvgn). From the CRISPResso2 (RRID:SCR_021538) reported 'Quantification_of_editing_frequency' table, the allele frequency of each group was calculated as follows:

Wild type (WT), ([Unmodified Reference] + [Only Substitution Reference])/[Total Reads aligned]
Pure primed editing (PPE), ([Unmodified Prime-edited] + [Only substitution Prime-edited])/[Total Reads aligned]
Impure primed editing (IPE), ([Total Prime-edited] – [Unmodified Prime-edited] – [Only substitution Prime-edited])/[Total Reads aligned]
By-product, 1-WT-PPE-IPE

## Software and statistics

Bar graphs were drawn in Graphpad Prism 9 (RRID:SCR_002798). Error bars indicate the SD. Number of biological replicates (N) is indicated in each figure legend. Heatmaps were generated using Morpheus (Broad Institute, RRID:SCR_017386).

## Acknowledgements

We would like to thank Devin Snyder for administrative support and help with editing of the manuscript. We thank Steven Poser for technical support, help with human hPSC culture and molecular biology. We thank Netravathi Krishnappa for providing NGS support. We thank all the members of the Hockemeyer, Soldner, Rio, Gilbert, and Bateup lab for helpful discussions and comments on the manuscript. We thank Sergiu P Pașca for providing the 8858 hiPSC line. This research was funded in part by Aligning Science Across Parkinson's ASAP-000486 through the Michael J Fox Foundation for Parkinson's Research (MJFF). Some of the Flow Cytometry and Genomics shared resources at Albert Einstein College of Medicine were supported by the Cancer Center Support Grant (P30 CA013330). HL is a fellow in the Siebel Stem Cell Institute. FS is supported by internal research support from the Department of Neuroscience at Albert Einstein College of Medicine. For the purpose of open access, the author has applied a CC BY public copyright license to all Author Accepted Manuscripts arising from this submission.

## Additional information

### Funding

| Funder | Grant reference number | Author |
|---|---|---|
| Aligning Science Across Parkinson's | ASAP-000486 | Luke A Gilbert<br>Donald C Rio<br>Dirk Hockemeyer<br>Frank Soldner |
| Albert Einstein College of Medicine, Yeshiva University | Internal research support from the Department of Neuroscience | Frank Soldner |
| National Cancer Institute | Cancer Center Core Support Grant to to the Albert Einstein College of Medicine Cancer Center; P30 CA013330 | Frank Soldner<br>Oriol Busquets |
| Siebel Stem Cell Institute | | Hanqin Li<br>Dirk Hockemeyer |
| Chan Zuckerberg Biohub | Investigators | Helen S Bateup<br>Dirk Hockemeyer |
| National Institutes of Health, Office of Strategic Coordination | DP2 CA239597 | Luke A Gilbert |
| National Human Genome Research Institute | R01 HG012227 | Luke A Gilbert |

The funders had no role in study design, data collection and interpretation, or the decision to submit the work for publication.

### Author contributions

Hanqin Li, Conceptualization, Data curation, Formal analysis, Validation, Investigation, Visualization, Methodology, Writing – original draft, Writing – review and editing; Oriol Busquets, Conceptualization, Data curation, Formal analysis, Validation, Investigation, Visualization, Writing – original draft, Writing – review and editing; Yogendra Verma, Gabriella R Pangilinan, Formal analysis, Investigation; Khaja Mohieddin Syed, Nitzan Kutnowski, Data curation, Formal analysis, Investigation, Writing – review and editing; Luke A Gilbert, Conceptualization, Funding acquisition, Writing – review and editing; Helen S Bateup, Supervision, Funding acquisition, Writing – review and editing; Donald C Rio, Conceptualization, Data curation, Formal analysis, Supervision, Funding acquisition, Validation, Investigation, Visualization, Methodology, Project administration, Writing – review and editing; Dirk Hockemeyer, Conceptualization, Data curation, Formal analysis, Supervision, Funding acquisition, Validation, Investigation, Visualization, Methodology, Writing – original draft, Project administration, Writing – review and editing; Frank Soldner, Conceptualization, Data curation, Formal analysis, Supervision, Funding acquisition, Validation, Investigation, Methodology, Writing – original draft, Project administration, Writing – review and editing

### Author ORCIDs

Hanqin Li  http://orcid.org/0000-0001-7995-1084
Oriol Busquets  http://orcid.org/0000-0002-1372-9699
Nitzan Kutnowski  http://orcid.org/0000-0002-3012-4616
Helen S Bateup  http://orcid.org/0000-0002-0135-0972
Donald C Rio  http://orcid.org/0000-0002-4775-3515
Dirk Hockemeyer  http://orcid.org/0000-0002-5598-5092
Frank Soldner  http://orcid.org/0000-0002-7102-8655

### Decision letter and Author response

Decision letter https://doi.org/10.7554/eLife.79208.sa1
Author response https://doi.org/10.7554/eLife.79208.sa2

## Additional files

### Supplementary files
• Supplementary file 1. Table providing DNA Oligonucleotides and gBlocks gene fragment sequences.
• Supplementary file 2. Table providing list of generated plasmids and synthetic gRNA/ngRNA/pegRNA sequences.
• MDAR checklist

### Data availability

Sequencing data can be accessed through the repository platform Zenodo (https://doi.org/10.5281/zenodo.6941502). The datasets for AAVS1 knock-in genotyping, aCGH karyotyping, and the source data files used to generate the featured graphs and tables can be found on Zenodo (https://doi.org/10.5281/zenodo.6941599). Plasmids referred to in this paper have been deposited to Addgene's Michael J. Fox Foundation Plasmid Resource and their associated RRID can be found in Supplemental table 2.

The following datasets were generated:

| Author(s) | Year | Dataset title | Dataset URL | Database and Identifier |
|---|---|---|---|---|
| Li H, Busquets O, Verma Y, Syed K, Kutnowski N, Pangilinan G, Gilbert L, Bateup H, Rio D, Hockemeyer D, Soldner F | 2022 | Bulk NGS/allele quantification - Highly efficient generation of isogenic pluripotent stem cell models using prime editing | https://doi.org/10.5281/zenodo.6941502 | Zenodo, 10.5281/zenodo.6941502 |
| Li H, Busquets O, Verma Y, Syed K, Kutnowski N, Pangilinan G, Gilbert L, Bateup H, Rio D, Hockemeyer D, Soldner F | 2022 | Highly efficient generation of isogenic pluripotent stem cell models using prime editing - Datasets | https://doi.org/10.5281/zenodo.6941599 | Zenodo, 10.5281/zenodo.6941599 |

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
