## [Editor Report]

In this manuscript, Li et al., directly compare different editing strategies for human pluripotent stem cells and demonstrate that prime editing is more efficient and precise, compared with double strand break-based methods. They also confirm the suitability of prime editing for the introduction of different mutations related to Parkinson’s disease as a model. In this process the authors noted a lower editing efficiency of human pluripotent stem cells, compared with tumour cell lines, and explored ways to improve it. Nucleofection of in vitro-transcribed mRNA-based delivery approach significantly increased the editing efficiency, without the need to select for targeted clones, and multiple rounds of mRNA-based prime editing could yield near complete editing of hPSCs, including disease-causing mutations. The proposed platform paves the way for future prime editing methods for hPSCs.

---

## [Decision Letter]

**Decision letter after peer review:**

Thank you for submitting your article “Highly efficient generation of isogenic pluripotent stem cell models using prime editing” for consideration by *eLife*. I apologise for the long time required for review. Your article has now been reviewed by 2 peer reviewers, one of whom is a member of our Board of Reviewing Editors, and the evaluation has been overseen by Mone Zaidi as the Senior Editor. The following individual involved in review of your submission has agreed to reveal their Identity: Angel Raya (Reviewer #2).

Essential revisions:

1) The authors should discuss the potential advantages and disadvantages of prime editing compared with other strategies, such as base editing. Although prime editing could target 90% human pathogenic variants and enables a wider range of sequencing changes, it has been associated with the generation of insertion and deletion base products as well.

2) Previous studies have reported undesired on-target edits, such as indels, using the PE3 system. The authors should discuss the reasons for their different results.

3) Other potential limitations, such as the complexity of pegRNA design and its effect on editing efficiency, should be discussed as well.

4) While the authors do search for unwanted editions in the vicinity of the targeted area, no attempt is made at detecting and quantifying more distant undesired editing events, including off-target events. This information should be important to readers when selecting a protocol for generating isogenic controls. The authors should better estimate the safety of their PE protocol. Probably the easiest way to do this would be by WGS of some selected edited versus parental iPSC lines. Particularly informative would be any of the multi-targeted cells shown in Figure 4A (if individual clones were obtained) or any of the SNCA (A30P)-corrected clones shown in Figure 4C, searching for de novo SNVs or indels potentially arising from off-target PE or from overall genotoxicity of the procedure (aCGH karyotyping is not resolutive enough for this). Analyzing genome stability of the clones generated in Figure 2 would also be informative to assess the safety of constitutively expressing nCas9-RT in iPSC.

5) The authors should discuss their findings in the light of a very similar study recently published by Habib et al., in NAR (PMID: 35018468).

6) Please correct the typo in page 20, line 450: 'For feeder feel culture,'.

*Reviewer #1 (Recommendations for the authors):*

The authors should discuss the potential advantages and disadvantages of prime editing compared with other strategies, such as base editing. Although prime editing could target 90% human pathogenic variants and enables a wider range of sequencing changes, it has been associated with the generation of insertion and deletion base products as well.

Previous studies have reported undesired on-target edits, such as indels, using the PE3 system. The authors should discuss the reasons for their different results.

Other potential limitations, such as the complexity of pegRNA design and its effect on editing efficiency, should be discussed as well.

*Reviewer #2 (Recommendations for the authors):*

I commend the authors for a well-designed and thorough study, which surely deserves being published. Below are some recommendations for improvement:

1. While the authors do search for unwanted editions in the vicinity of the targeted area, no attempt is made at detecting and quantifying more distant undesired editing events, including off-target events. This information should be important to readers when selecting a protocol for generating isogenic controls. The authors should better estimate the safety of their PE protocol. Probably the easiest way to do this would be by WGS of some selected edited versus parental iPSC lines. Particularly informative would be any of the multi-targeted cells shown in Figure 4A (if individual clones were obtained) or any of the SNCA (A30P)-corrected clones shown in Figure 4C, searching for de novo SNVs or indels potentially arising from off-target PE or from overall genotoxicity of the procedure (aCGH karyotyping is not resolutive enough for this). Analyzing genome stability of the clones generated in Figure 2 would also be informative to assess the safety of constitutively expressing nCas9-RT in iPSC.

2. The authors should discuss their findings in the light of a very similar study recently published by Habib et al., in NAR (PMID: 35018468).

3. Please, edit typo in page 20, line 450: 'For feeder feel culture,'.

---

## [Author Response]

Essential revisions:1) The authors should discuss the potential advantages and disadvantages of prime editing compared with other strategies, such as base editing. Although prime editing could target 90% human pathogenic variants and enables a wider range of sequencing changes, it has been associated with the generation of insertion and deletion base products as well.

Based on the reviewers’ suggestions, we have included the following paragraph discussing PE in the context of other genome editing approaches:

“This work focuses on PE, which is highly versatile to introduce not only a wide range of disease-associated single nucleotide sequence variants but also more complex genetic alterations such as insertions and deletions. However, there are additional non-DSB-based genome editing approaches (e.g., based editors) which, dependent on the specific context, have been shown to efficiently introduce genetic modifications in hPSCs we refer to recent reviews for a detailed discussion of advantages and disadvantages of such approaches (Anzalone et al., 2020; Molla et al., 2021; Zeballos and Gaj, 2021). Although not tested, we believe that the here described RNA transfection-based delivery modalities could be adapted to increase genome editing efficiencies for genome editing approaches other than PE.”

2) Previous studies have reported undesired on-target edits, such as indels, using the PE3 system. The authors should discuss the reasons for their different results.

In the discussion of our revised manuscript, we specifically address undesired on-target edits (Comment 2) in the context of the recently published study published by Habib et al., in NAR (PMID: 35018468). Comparable to previous reports, we observed to a certain degree undesired on-target edits in particular when using the PE3 approach. As has been shown before, undesired edits are locus and pegRNA design-dependent and vary widely across genomic loci (Anzalone *et al.*, 2019; Chen *et al.*, 2021; Nelson *et al.*, 2021). While the frequency of undesired on-target modifications in our work is low compared to the intended sequence modifications, our study was not designed to allow for a quantitative comparison to previous experiments. We therefore refrain from specifically discussing such differences; however, we point out that the frequency of undesired edits might be affected by the different delivery modalities (e.g., mRNA transfection of the PE component (nCas9-RT) paired with the transfection of chemically-modified guide RNAs has different kinetics and expression levels compared to other approaches). Due to the unique property of hPSCs to allow for the expansion of clonal cell lines, we do not believe that such on-target indels limit the use of PE for disease modeling approaches because undesired modifications can be easily excluded through targeted sequencing during quality control of individual clones. Based on the reviewers’ suggestions, we have added the following paragraph to discuss undesired on-target edits:

“A recent study reported comparable high PE efficiencies in a doxycycline-inducible PE2-expressing hESC line (Habib *et al.*, 2022). The authors observe that PE3-mediated prime editing is generally accompanied by the generation of indels around the target site cause by the combinatory activity of the RT and pegRNA. While we also find a certain degree of indels (IPE and by-products) at the target site using the PE3 approach, the frequency is usually low compared to the intended sequence modifications which is consistent with previous data in other cell types (Anzalone *et al.*, 2019; Chen *et al.*, 2021; Nelson *et al.*, 2021). While it has been shown that the frequency of undesired edits varies widely across genomic loci and is pegRNA dependent, it is possible that indel frequency is also affected by the different delivery modalities (e.g., different mRNA transfection of the PE component paired with the transfection of chemically-modified guide RNAs has different kinetics and expression levels compared to other approaches). Due to the unique property of hPSCs to allow for the expansion of clonal cell lines, we do not believe that such on-target indels limit the use of PE for disease modeling approaches because undesired modifications can be easily excluded through targeted sequencing during quality control of individual clones.”

3) Other potential limitations, such as the complexity of pegRNA design and its effect on editing efficiency, should be discussed as well.

Based on the reviewers’ suggestions, we modified the following paragraph in the discussion to include potential limitations related to complexity of pegRNA design. The new paragraph reads:

“As it is the case for all genome editing approaches, PE efficiencies vary widely depending on the specific genomic context and pegRNA design and certain genetic modifications will require more extensive pegRNA testing and validation. However we expect that systematic approaches that establish optimized design parameters for prime editing, as recently described for cancer cells (Kim et al., 2020b; Nelson *et al.*, 2021) and the development of Cas9 variants with non-classical PAMs (Chatterjee et al., 2020; Kleinstiver et al., 2015; Miller et al., 2020) will overcome these limitations and combined with the optimized protocols reported here will allow PE to become a general method of choice for genome editing in hPSCs.”

4) While the authors do search for unwanted editions in the vicinity of the targeted area, no attempt is made at detecting and quantifying more distant undesired editing events, including off-target events. This information should be important to readers when selecting a protocol for generating isogenic controls. The authors should better estimate the safety of their PE protocol. Probably the easiest way to do this would be by WGS of some selected edited versus parental iPSC lines. Particularly informative would be any of the multi-targeted cells shown in Figure 4A (if individual clones were obtained) or any of the SNCA (A30P)-corrected clones shown in Figure 4C, searching for de novo SNVs or indels potentially arising from off-target PE or from overall genotoxicity of the procedure (aCGH karyotyping is not resolutive enough for this). Analyzing genome stability of the clones generated in Figure 2 would also be informative to assess the safety of constitutively expressing nCas9-RT in iPSC.

As already explained in detail above, we agree with the reviewers that an off-target analysis would “be important to the readers when selecting a protocol for generating isogenic controls.” However, we believe that there are already available substantial datasets in a variety of human cell-types indicating that Prime Editing (PE) is highly specific and shows much lower sgRNA-dependent or sgRNA-independent off-target effects compared to double-strand break (DSB)-based CRISPR/Cas9 or base editing approaches (Gao et al., 2021; Geurts et al., 2021; Habib *et al.*, 2022; Jin et al., 2021; Kim et al., 2020a; Schene et al., 2020). In our opinion, genome-wide off-target analyses, including whole genome sequencing of the vitro generated cell lines, as suggested by the reviewers, would be confounded by the substantial number of genetic alterations which occur during regular cell culture of human pluripotent stem cells (hPSCs). Recent analyses of clonal hPSC mutation estimates, indicate that individual cells lines acquire 3.5 base substitutions per population doubling, which amounts to hundreds of mutations between any clonal cell line after only a few weeks in culture (Kuijk et al., 2020). Therefore, it becomes impossible to distinguish in vitro culture mutations from PE-induced changes by simple whole genome sequencing of the correctly targeted clones. We believe that even sequencing all targeted clones and a comparable number of non-targeted subclones (which we would first have to generate) would not have sufficient statistical power to even detect common off-target sites. In our opinion, this analysis would be far beyond the scope of this paper. The key new finding here is the comparison of multiple delivery methods for PE reagents showing that synthetic mRNA is superior to other methods. This result should be of broad interest to scientific community interested in disease modeling in hPSCs.

It should be emphasized again that the focus of this work is to describe a novel and highly efficient platform for PE of hPSCs. We feel that based on the issues outlined above, a detailed off-target analysis, as suggested by the reviewers is not going to add much to the main findings in the current manuscript, since we are not using this method for human therapeutic editing. These experiments would be extremely complex, time consuming and require substantial resources and probably not add novel insights beyond what is already described in the literature about off-target modification of PE. While phenotypic effects resulting from off-target modifications in the context of disease modeling can and should be easily managed by simply analyzing multiple independently derived cell lines, a key future step towards the development of clinical prime editing approaches will require the development of sophisticated off-target analyses tools that account for nCAS9 single strand break-mediated and transient reverse transcriptase expression-mediated genetic alterations. Based on the reviewers’ comments, we have therefore added an additional paragraph to specifically discuss off-target modifications and why we didn’t include such an analysis in our manuscript. Again, we note that many previous published studies have shown that PE has many fewer “off-target” edits than either classic Cas9 editing and dCas9-base editing. The new paragraph reads:

“The focus of this work was to establish a highly efficient platform for PE in hPSCs and although all generated prime edited cell lines showed a normal karyotype consistent with low genotoxicity of PE due to the lack of DSBs (Anzalone *et al.*, 2019), it is important to point out that we refrained from a detailed off-target analysis. We believe that there are already available substantial datasets in a variety of cell-types indicating that PE is highly specific and shows much lower sgRNA-dependent or independent off-target effects compared to DSB-based CRISPR/Cas9 or base editing approaches (Gao et al., 2021; Geurts et al., 2021; Habib *et al.*, 2022; Jin et al., 2021; Kim et al., 2020a; Schene et al., 2020). Genome-wide off-target analyses of in vitro generated cell lines including whole genome sequencing-based approaches remain challenging even for conventional CRISPR/Cas9-based cutting approaches due to the substantial number of genetic alterations which occur during regular cell culture (Kuijk et al., 2020). Thus, a key future step towards the development of clinical prime editing approaches will require the development of sophisticated off-target analyses tools that account for nCAS9 single strand break-mediated and transient reverse transcriptase expression-mediated genetic alterations.”

5) The authors should discuss their findings in the light of a very similar study recently published by Habib et al., in NAR (PMID: 35018468).

We apologize for not discussing this recent paper in our initial submission. We have now included a discussion of this paper in our revised manuscript (see combined response above for reviewer comments 2 and 5).

6) Please correct the typo in page 20, line 450: 'For feeder feel culture,'.

We thanks the reviewer and have corrected the typo.